# Dynamics of sputum conversion during effective tuberculosis treatment: A systematic review and meta-analysis

Claire J. Calderwood[1☯], James P. Wilson[1☯], Katherine L. Fielding[1], Rebecca C. Harris[1], Aaron S. Karat[1], Raoul Mansukhani[1], Jane Falconer[2], Malin Bergstrom[1], Sarah M. Johnson[1], Nicky McCreesh[1], Edward J. M. Monk[1], Jasantha Odayar[3], Peter J. Scott[1], Sarah A. Stokes[1], Hannah Theodorou[1], David A. J. Moore[1]*

**1** TB Centre, London School of Hygiene & Tropical Medicine, London, United Kingdom, **2** Library & Archives Service, London School of Hygiene & Tropical Medicine, London, United Kingdom, **3** Division of Epidemiology and Biostatistics, School of Public Health & Family Medicine, University of Cape Town, Cape Town, South Africa

☯ These authors contributed equally to this work.
* david.moore@lshtm.ac.uk

**Data Availability Statement:** Data are available from the primary studies, all of which are published.

**Funding:** This work was partly supported by funding from the MRC (ref: MR/K012126/1), and also by a contract to the LSHTM TB Centre from

## Abstract

### Background

Two weeks' isolation is widely recommended for people commencing treatment for pulmonary tuberculosis (TB). The evidence that this corresponds to clearance of potentially infectious tuberculous mycobacteria in sputum is not well established. This World Health Organization–commissioned review investigated sputum sterilisation dynamics during TB treatment.

### Methods and findings

For the main analysis, 2 systematic literature searches of OvidSP MEDLINE, Embase, and Global Health, and EBSCO CINAHL Plus were conducted to identify studies with data on TB infectiousness (all studies to search date, 1 December 2017) and all randomised controlled trials (RCTs) for drug-susceptible TB (from 1 January 1990 to search date, 20 February 2018). Included articles reported on patients receiving effective treatment for culture-confirmed drug-susceptible pulmonary TB. The outcome of interest was sputum bacteriological conversion: the proportion of patients having converted by a defined time point or a summary measure of time to conversion, assessed by smear or culture. Any study design with 10 or more particpants was considered. Record sifting and data extraction were performed in duplicate. Random effects meta-analyses were performed. A narrative summary additionally describes the results of a systematic search for data evaluating infectiousness from humans to experimental animals (PubMed, all studies to 27 March 2018). Other evidence on duration of infectiousness—including studies reporting on cough dynamics, human tuberculin skin test conversion, or early bactericidal activity of TB treatments—was outside the scope of this review. The literature search was repeated on 22 November 2020, at the request of the editors, to identify studies published after the previous censor date. Four small studies reporting 3 different outcome measures were identified, which included

the Global TB Programme of the World Health Organization (WHO) (ref: 2017/748478-0). The work was commissioned by the WHO Global TB Department Guideline Development Group (GDG) who defined the scope. The study methodology was developed by the authors in conjunction with members of the GDG. The WHO played no role in the data collection, data analysis, manuscript preparation or decision to publish.

**Competing interests:** The authors have declared that no competing interests exist.

**Abbreviations:** DS-TB, drug-susceptible pulmonary tuberculosis; GDG, Guideline Development Group; IPC, infection prevention and control; MDR-TB, multi-drug-resistant tuberculosis; *Mtb*, *Mycobacterium tuberculosis*; PC, proportion converted; PTB, pulmonary tuberculosis; RCT, randomised controlled trial; TB, tuberculosis; TST, tuberculin skin test; TTC, time to conversion; WHO, World Health Organization.

no data that would alter the findings of the review; they are not included in the meta-analyses. Of 5,290 identified records, 44 were included. Twenty-seven (61%) were RCTs and 17 (39%) were cohort studies. Thirteen studies (30%) reported data from Africa, 12 (27%) from Asia, 6 (14%) from South America, 5 (11%) from North America, and 4 (9%) from Europe. Four studies reported data from multiple continents. Summary estimates suggested smear conversion in 9% of patients at 2 weeks (95% CI 3%–24%, 1 single study [$N = 1$]), and 82% of patients at 2 months of treatment (95% CI 78%–86%, $N = 10$). Among baseline smear-positive patients, solid culture conversion occurred by 2 weeks in 5% (95% CI 0%–14%, $N = 2$), increasing to 88% at 2 months (95% CI 84%–92%, $N = 20$). At equivalent time points, liquid culture conversion was achieved in 3% (95% CI 1%–16%, $N = 1$) and 59% (95% CI 47%–70%, $N = 8$). Significant heterogeneity was observed. Further interrogation of the data to explain this heterogeneity was limited by the lack of disaggregation of results, including by factors such as HIV status, baseline smear status, and the presence or absence of lung cavitation.

## Conclusions

This systematic review found that most patients remained culture positive at 2 weeks of TB treatment, challenging the view that individuals are not infectious after this interval. Culture positivity is, however, only 1 component of infectiousness, with reduced cough frequency and aerosol generation after TB treatment initiation likely to also be important. Studies that integrate our findings with data on cough dynamics could provide a more complete perspective on potential transmission of *Mycobacterium tuberculosis* by individuals on treatment.

## Trial registration

**Systematic review registration:** PROSPERO 85226.

---

## Author summary

### Why was this study done?

- Though cited in many countries' national guidance, the evidence that individuals with pulmonary tuberculosis (TB) are rendered non-infectious by 2 weeks of effective TB treatment is challenged.

- This systematic review was commissioned by the World Health Organization to provide evidence to inform TB infection prevention and control guidelines.

- We sought to synthesise the available data on the clearance of potentially infectious TB bacteria from the sputum of patients after starting effective treatment.

### What did the researchers do and find?

- We performed systematic searches of literature databases to identify relevant articles, using predetermined inclusion criteria. Extracted data were synthesised using narrative summaries and meta-analyses.

- A minority of patients had clearance of TB bacteria from sputum at 2 weeks of effective treatment, as assessed by either sputum smear or culture.

- As expected, the proportion having cleared TB bacteria from sputum increased over time; however, at 2 months of treatment 12% and 41% of patients still had viable TB bacteria present, as assessed by solid and liquid culture, respectively.

### What do these findings mean?

- The presence of viable TB bacteria in the sputum of pulmonary TB patients beyond 2 weeks of effective treatment suggests individuals may be infectious for longer than this interval.

- TB transmission requires the presence of viable mycobacteria in sputum and a mechanism for aerosol or droplet spread. Understanding how other factors, such as the presence of cough, change during treatment is also important for TB infection prevention and control.

## Introduction

Tuberculosis (TB) is the leading cause of death from an infectious disease [1]. *Mycobacterium tuberculosis (Mtb)*, the causative organism, is transmitted via respiratory droplet nuclei generated by individuals with pulmonary TB (PTB). Transmission is affected by environmental factors, characteristics of both the source case and exposed individual, and the nature of their contact. The greatest risk is from individuals with smear-positive PTB and high cough frequency [2–4]. Interruption of transmission requires identification and separation of infectious individuals until they are rendered non-infectious through treatment. Multiple countries' national guidance documents recommend isolation of hospitalised individuals, with 2 weeks of effective treatment commonly cited as the time frame after which patients are considered non-infectious [5–9]. Implementing respiratory isolation has considerable resource implications, and is often unachievable in overstretched health systems in low- and middle-income countries, where the greatest burden of TB lies [1].

The evidence base for the 2-week 'rule' has been repeatedly challenged by data demonstrating that most baseline sputum-culture-positive patients positive remain so for longer than 14 days [10]. These individuals are potentially infectious, although diminished cough frequency may reduce transmission [4,11].

This systematic review was commissioned by the World Health Organization (WHO) Department of Global TB Programme Guideline Development Group (GDG) to synthesise evidence on the dynamics of potential infectiousness of individuals with PTB receiving effective therapy, informing the 2019 guideline update on TB infection prevention and control (IPC) [12].

Potential infectiousness was defined here as detection of *Mtb* by sputum smear or culture, irrespective of cough dynamics or contact mixing patterns. Whilst smear microscopy may detect non-viable, and therefore non-infectious, bacilli, it is referenced as a measure of treatment response in published guidelines and in clinical practice. Molecular methods (*Mtb* DNA detection, e.g., Xpert MTB/RIF [Cepheid]) also detect non-viable bacilli and are not routinely used in assessment of treatment response; such assessments were excluded from this analysis.

Two types of evidence were considered: (1) data on time from treatment initiation to sputum smear and culture conversion and (2) data from measurements of infectiousness from PTB patients to experimental animals. Although cough is clearly important, review of cough dynamics during TB treatment did not form part of this study. Studies evaluating early bactericidal activity and human-to-human transmission were also considered outside the scope of this study.

## Methods

This review is reported following Preferred Reporting Items for Systematic Reviews and Meta-Analyses (PRISMA) guidelines [13]. Our PRISMA checklist is located in S1 PRISMA Checklist. A prospectively registered review protocol is available in PROSPERO (https://www.crd.york.ac.uk/PROSPERO, study ID 85226 [background question 3]).

### Search strategy

A professional librarian developed and executed the literature search, in consultation with the WHO GDG. The population of interest was patients receiving an effective treatment regimen for drug-susceptible PTB (DS-TB); the outcome of interest was bacteriological conversion of sputum. Effective treatment required at least rifampicin (R) and isoniazid (H) throughout, pyrazinamide (Z) for the 2-month intensive phase, and administration at least 5 days per week. Inclusion of ethambutol (E) and streptomycin (S) was permitted but not essential. Regimens including drugs not conventionally regarded as first-line, such as quinolones, were excluded [14]. The outcome was defined as either the proportion of participants achieving bacteriological (smear or culture) conversion from positive to negative by fixed time points during treatment (proportion converted [PC]) or time to conversion (TTC) for the study population. As a descriptive analysis, no intervention or comparator group was defined.

Full search parameters are detailed in S1 Table and S2 Table. The search was constructed in OvidSP MEDLINE, adapted and run in OvidSP Embase; OvidSP Global Health; and EBSCO CINAHL Plus. Searches used subject headings where available, and search terms in the title and abstract. Combinations of terms were used to capture the concepts of infectiousness and TB. Language was limited to English, Japanese, Chinese, Russian, French, Spanish, and Portuguese. On initial screening of search results, the authors noted the absence of several randomised controlled trials (RCTs) known to include culture conversion data in standard treatment arms. An additional search was therefore constructed identifying all RCTs of DS-TB treatment in humans. The first search was conducted on 1 December 2017 without date restrictions; the second (RCT search) was conducted on 20 February 2018, with date limited to 1990 onwards, to identify RCTs incorporating standard daily rifampicin-based regimens.

To mitigate the risk of overlooking important data reported since our search censor date, during manuscript review we conducted a further literature search of OvidSP MEDLINE and Embase on 22 November 2020. We used the original search terms (S1 Table and S2 Table) combined with terms identified as being common either as keywords or in the titles/abstracts of studies already included in the review. Full details, including additional search terms, are provided in S1 Appendix.

### Study selection and data extraction

Two-stage sifting by 2 independent reviewers was employed, applying predetermined eligibility criteria at title and abstract review and, where necessary, full-text screening. References and citations for all included studies were reviewed to identify further relevant articles. Inclusion and exclusion criteria are defined in Table 1.

**Table 1. Summary of inclusion and exclusion criteria employed during record sifting.**

| Component | Inclusion criteria | Exclusion criteria |
|---|---|---|
| Participants | • Bacteriologically confirmed (smear- or culture-positive) pulmonary tuberculosis<br>• Confirmed drug susceptibility to, at least, rifampicin and isoniazid[a]<br>• Treated with a regimen including rifampicin and isoniazid with pyrazinamide added during 2-month 'intensive phase', administered at least 5 days per week[b] | • Any study not reporting data on participants with confirmed, drug-susceptible pulmonary tuberculosis (or disaggregated data for this subgroup)<br>• Any study not in humans<br>• Any study reporting on participants not treated with a conventional first-line regimen [14] (e.g., addition of quinolones), or with treatment given fewer than 5 days per week[b] |
| Outcome measures | Studies reporting data on at least 1 of the outcome measures of interest:<br>• Proportion of participants achieving sputum smear microscopy or culture conversion[c], assessed at any time point within the first 4 months of treatment<br>• A summary measure (such as mean or median) of time to conversion, assessed by repeated smear or culture[c] assessment (at any interval) during treatment | • Any study not reporting on any of the outcomes of interest |
| Study types | • Any consecutive case series, case–control study, cohort study, randomised controlled study, systematic review, or meta-analysis | • Any systematic review superseded by an updated systematic review<br>• Narrative reviews not adding new data or new analysis of data to existing knowledge<br>• Commentaries and mathematical modelling studies<br>• Studies with fewer than 10 participants per arm<br>• Any study not written in English, Japanese, Chinese, Russian, French, Spanish, or Portuguese<br>• Any study published before 1946 |

[a]Criterion only applied at full-text review, as infrequently reported in abstracts.

[b]Where studies used an included regimen for an initial period, with an excluded regimen later (e.g., during continuation phase, or insufficient duration of pyrazinamide), the study was included and data extracted only for the period where an included regimen was administered.

[c]Sputum smear and culture assessment could be by any method including Ziehl–Neelsen stain, fluorescence microscopy, liquid or solid culture, or an undefined culture method.

Data were extracted in duplicate into a standardised, pilot-tested Excel database (Microsoft Office 2016) (S1 Form). As our intention was to analyse an 'on treatment' population, data from per-protocol analyses were preferentially used. Data were not directly extracted from systematic reviews; instead, source papers were identified and reviewed for inclusion. At each stage of screening and extraction, disagreements were resolved by a third independent reviewer.

Quality assessment was performed by 2 independent reviewers at study level using an adapted National Institutes of Health tool for case series [15].

## Analysis

Extracted data were synthesised using a narrative approach, and meta-analyses where appropriate. TTC data were presented as study-specific estimates (mean or median) together with a measure of spread (interquartile range, standard error, standard deviation, or range) where provided.

Random effects meta-analyses were used to calculate pooled conversion proportions across detection method subgroups (smear, solid culture, or liquid culture) at different time points. Where the culture type was not stated, those performed prior to 1990 were assumed to use solid media; others are reported as 'unspecified'. Study-specific CIs were calculated using the score method. Weighted pooled proportion estimates were calculated for each subgroup using the Freeman–Tukey double arcsine transformation, avoiding potential bias introduced from excluding studies with conversion proportions equalling 0 or 1 [16]. Pooled estimate CIs were calculated using the Wald method. Within-subgroup heterogeneity was assessed with the chi-

squared test and described with the $I^2$ statistic. Data cleaning was conducted in Stata (Stata/IC version 13.1), and meta-analyses performed in R (R Core Team, 2020) using the 'forest.meta' function from the package 'meta' [17].

### Animal studies

A review of data on *Mtb* transmission from humans to animals was explicitly requested by the WHO GDG and was undertaken as a separate search and analysis. The population of interest was patients receiving effective treatment for drug-susceptible or drug-resistant PTB. The outcome of interest was transmission to experimental animals exposed to air exhausted from isolation facilities accommodating TB patients, defined as tuberculin skin test (TST) conversion or new active TB disease in these animals.

The search was run on 27 March 2018; details are provided in S3 Table. Titles and abstracts, then full texts of identified papers were reviewed. References and citations of included papers were also reviewed. The diversity and heterogeneity of methodologies and analyses applied by included studies did not permit standardised data extraction or quality assessment. Relevant data were extracted for narrative synthesis by 1 reviewer and independently checked for accuracy by a second.

## Results

In total, 5,290 unique records were identified for title and abstract review (first search: 3,558; RCT search: 1,732). Full-text review of 180 papers identified 22 studies for inclusion in the main analysis. A further 22 articles were added from references and citations (Fig 1), resulting in 44 papers for analysis (Table 2).

The further literature search conducted in November 2020 identified 4 small studies meeting the inclusion criteria, each with under 150 participants and reporting 3 different outcome measures [18–21]. The results reported in these studies did not address any of the data gaps in important subgroups and had no material impact upon the results or conclusions of this review so were not incorporated into the meta-analyses. Further details of the eligible studies are provided in Table B in S1 Appendix.

Of the included papers, 27 (61%) were RCTs and 17 (39%) were cohort studies (Table 2). Thirteen studies (30%) reported data from Africa, 12 (27%) from Asia, 6 (14%) from South America, 5 (11%) from North America, and 4 (9%) from Europe. Four studies reported data from multiple continents. Intensive phase treatment in most studies comprised RHZE (31 studies), 6 studies used streptomycin instead of ethambutol, and the remainder used other acceptable regimens (RHZ with or without E or S). Seventeen studies included people living with HIV, representing between 0.8% and 61% of the study population. Thirty-three studies only included people with baseline smear-positive PTB.

Eight studies reported both PC and TTC data, 31 studies reported only PC data, and 5 reported only TTC data. For data on the proportion of patients achieving bacteriological conversion, there were a total of 96 estimates: 4, 5, and 6 estimates at 1, 2, and 3 weeks of treatment, respectively, and 20, 46, 7, and 8 estimates at 1, 2, 3, and 4 months, respectively (Table 3).

Thirty-seven studies were ranked as 'good' in the quality scoring system ($\geq$7/9), with 7 studies ranked as 'fair' (4–6/9), and none as 'poor' quality (S4 Table).

### Smear conversion

Fifteen studies reporting data on smear conversion were included, providing 18 estimates of the proportion of individuals who had smear converted at defined time points on treatment and 6 summary estimates of the time to smear conversion.

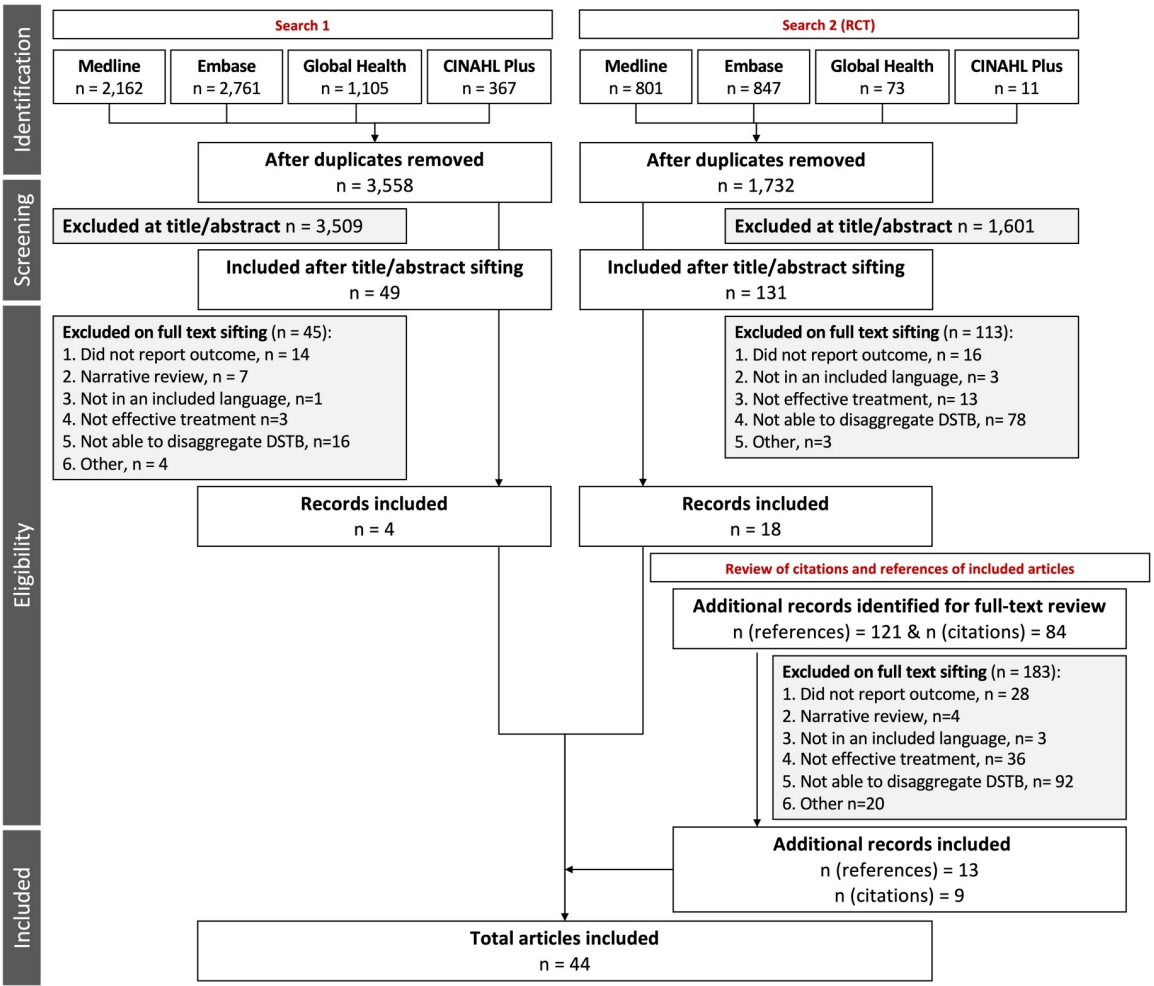

**Fig 1. Flow diagram showing databases searched and process for article inclusion.** Two searches were developed, conducted by a professional librarian: a primary search and a second search that aimed to identify all RCTs of tuberculosis treatment conducted since 1990, as described in text. DSTB, drug-susceptible tuberculosis; n, number of studies; RCT, randomised controlled trial.

For the 6 studies reporting a summary estimate of time to smear conversion, the median TTC ranged from 20 to 27 days and the mean TTC ranged from 29 to 55 days (Table 4). Shorter times to conversion were seen in studies with more frequent sampling.

The proportions of treated patients with baseline smear-positive DS-TB achieving positive-to-negative smear conversion are shown in Fig 2. From a total of 18 estimates across 11 studies, the proportions of baseline smear-positive patients achieving smear conversion at the 1-, 2-, 3-, and 4-month time points were 33% (95% CI 25%–42%), 82% (95% CI 78%–86%), 94% (95% CI 94%–95%), and 100% (95% CI 72%–100%), respectively.

## Culture conversion

Thirty-seven studies reported data on culture conversion. Ten studies published after 1990 did not report the culture method and were excluded from the main analysis (TTC and PC data with meta-analysis are reported in S5 Table and S1 Fig, respectively.).

Among studies with data on culture method, 10 used liquid culture, 25 used solid, and 8 used both. Summary statistics for TTC were reported in 11 studies. These are summarised by

**Table 2. Description of included studies.**

| Study | Year | Location | Study design | N assessed* | Age (years)† | Percent female | Percent HIV+ | Treatment‡ Intensive | Continuation | Adherence support | Outcome reported (PC or TTC) | Assessment method§ | Study dates |
|---|---|---|---|---|---|---|---|---|---|---|---|---|---|
| **Smear-positive pulmonary TB** | | | | | | | | | | | | | |
| Abal [22] | 2005 | Kuwait | C (p) | 339 | — | Smoker −: 42%; Smoker +: 1% | — | 2RHZ±E/S | — | — | PC | Smear (ZN) | 1998–2000 |
| Chaulet [23] | 1995 | Algeria | RCT | 196 | — | 26% | — | 2RHZ | 4RH | IP DOT (≥14 d)/ U-INH | PC | Culture (NS) | — |
| Conde [24] | 2009 | Brazil | RCT | 72 | 36 (±12) | 31% | 3% | 2RHZE (5 d) | 4RH | DOT | Both | Culture (solid) | 2004–2007 |
| Conde [25] | 2016 | Brazil | RCT | 45 [59] | 30 (24–45) | 39% | 0% | 2RHZE | 4RH | DOT | Both | Culture (solid + liquid) | 2009–2013 |
| Dawson [26] | 2009 | South Africa | RCT | 30 | 32 (±11) | 37% | 7% | 2RHZE | 4RH¶ | DOT | PC | Smear (NS); culture (NS) | 2005–2006 |
| Dawson [27] (2015a) | 2015 | South Africa, Tanzania | RCT | 54 [59] | 30.4 (±10) | 31% | 22% | 2RHZE | — | — | Both | Culture (solid + liquid) | 2012–2013 |
| Dawson [28] (2015b) | 2015 | South Africa | RCT | 36 [48] | 30 (25–36) | 27% | 6% | 2RHZE | 4RH | DOT‖ | Both | Culture (solid + liquid) | 2010–2013 |
| Desjardin [29] | 1999 | Brazil | RCT | 19 | 34 (range 19–55) | 26% | 0% | 2RHZE | — | DOT (≥2 wk)/pill count/ U-INH | PC | Culture (solid) | — |
| Dlugovitzky [30] | 2006 | Argentina | RCT | 10 | 37 (±13) | 20% | 0% | 2RHZE | 4RH | DOT | PC | Smear (ZN); culture (solid) | ≥2001 |
| Dominguez-Castellano [31] | 2003 | Spain | C (p) | 95 [109] | — | — | 39% | 2RHZ±E | NS | — | TTC | Smear (ZN) | — |
| Dorman [32] | 2009 | Multiple (4 continents) | RCT | 164 | 30 (25–40) | 28% | 11% | 2RHZE (5 d) | — | DOT‖ | PC | Culture (solid + liquid) | 2006–2007 |
| Dorman [33] | 2012 | Multiple (4 continents) | RCT | 183 | 34 (26–47) | 37% | 13% | 2RHZE (5 d) | — | DOT‖ | PC | Culture (solid + liquid) | 2008–2010 |
| Dorman [34] | 2015 | Multiple (4 continents) | RCT | 64 [85] | 33 (19–78) | 35% | 6% | 2RHZE | — | DOT | PC | Culture (solid + liquid) | 2011–2012 |
| ECA/BMRC [35] | 1983 | Multiple (SSA) | RCT | 708 | — | 35% | — | 2RHZS | — | IP DOT | PC | Smear (NS) | 1978–1980 |
| Grandjean [36] | 2015 | Peru | C (p) | 487 | 29 | 61% | 4% | 2RHZE¶ | — | — | PC | Smear (NS) | 2010–2013 |
| HKCS/ BMRC [37] | 1981 | Hong Kong | RCT | 168 [1,207] | — | 28% | — | 6RHZE | — | DOT‖ | PC | Culture (solid) | — |
| HKCS/ BMRC [38] | 1978 | Hong Kong | RCT | 180 [842] | — | 29% | — | 2RHZS | — | DOT | PC | Culture (solid) | — |
| Jindani [39] | 2014 | Multiple (SSA) | RCT | 163 | — | 36% | 29% | 2RHZE | — | DOT | PC | Culture (solid + liquid) | 2008–2011 |

*(Continued)*

**Table 2.** (Continued)

| Study | Year | Location | Study design | N assessed* | Age (years)† | Percent female | Percent HIV+ | Treatment‡ | | Adherence support | Outcome reported (PC or TTC) | Assessment method§ | Study dates |
|---|---|---|---|---|---|---|---|---|---|---|---|---|---|
| | | | | | | | | Intensive | Continuation | | | | |
| Jindani [40] | 2016 | Multiple (SSA, S. Am.) | RCT | 92 [100] | 29 (range 18–67) | 34% | 0% | 2RHZE | — | DOT | PC | Culture (solid) | 2011–2013 |
| Johnson [41] | 2000 | Uganda | RCT | 52 [59] | 29 (±8) | 22% | 0% | 2RHZE | 4RH | Pill count/ U-INH | PC | Culture (solid) | 1995–1997 |
| Johnson [42] | 2003 | Uganda | RCT | 47 [55] | 27.4 (±7) | 25% | 0% | 2RHZE | — | IP DOT (≥1 mo)/ U-INH | PC | Culture (solid) | — |
| Joloba [43] | 2000 | Uganda | C (p) | 36 | HIV−: 31 (±9); HIV+: 27 (±8) | HIV−: 57%; HIV+: 50% | 61% | 2RHZE | — | DOT (≥2 wk) | PC | Culture (solid + liquid) | 1996–1997 |
| Kanda [44] | 2015 | Japan | C (r) | 86 | 57 (range 40–67) | 30% | 0% | 2RHZ+E/S | ≥4RH | IP DOT | Both | Culture (solid) | — |
| Kennedy [45] | 1996 | Tanzania | RCT | 86 | 36 (±13) | 36% | 37% | 2RHZE | 2RHZ+2RH | IP DOT | TTC | Smear (FM); culture (NS) | 1990–1992 |
| Long [46] | 2003 | Canada | C (p/ r) | 32 | — | 38% | 0% | 2RHZ | — | DOT‖ | Both | Smear (ZN) | 1997–1999 |
| Méchaï [47] | 2016 | France | C (p) | 30 | 43 (36–52) | 23% | 0% | 2RHZE | 4RH | — | TTC | Smear (NS); culture (NS) | 2009–2013 |
| Pheiffer [48] | 2008 | South Africa | C (p) | 105 | — | — | — | 2RHZE | — | — | PC | Smear (ZN) | 2000–2004 |
| Rathored [49] | 2012 | India | C (p) | 50 [338] | 27 (±10) | 28% | 0% | 2RHZE¶ | 2RH¶ | DOT | TTC | Smear (ZN); culture (solid) | 2006–2011 |
| STBS/BMRC [50] | 1979 | Singapore | RCT | 330 [363] | — | 35% | — | 2RHZS | — | DOT | PC | Culture (solid) | — |
| STBS/BMRC [51] | 1985 | Singapore | RCT | 319 | — | 40% | — | 1RHZS/ 2RHZS/ 2RHZ | — | DOT | PC | Culture (solid) | — |
| Singla [52] | 2003 | Saudi Arabia | C (r) | 434 [514] | Male: 38 (±33); female: 33 (±15) | 37% | 0% | 2RHZE | — | IP DOT (≥1 mo) | PC | Smear (NS) | 1998–1999 |
| Stoffel [53] | 2014 | Argentina | C (r) | 148 | — | 34% | — | 2RHZE | — | DOT | PC | Smear (ZN); culture (solid) | 2000–2010 |
| Tanzania/ BMRC [54] | 1985 | Tanzania | RCT | 224 | — | 29% | — | 2RHZS | — | IP DOT/ U-INH | PC | Culture (solid) | 1978–1981 |
| Tanzania/ BMRC [55] | 1996 | Tanzania | RCT | 266 | — | 31% | — | 1.5RHZS | — | IP DOT/ U-INH | PC | Culture (NS) | 1982–1985 |
| Telzak [56] | 1997 | US | C (r) | 77 [100] | — | 39% | 59% | RHZE¶,** | RH‖ | DOT | TTC | Smear (ZN); culture (NS) | 1993–1996 |
| **Any pulmonary TB** | | | | | | | | | | | | | |
| BTA [57] | 1981 | UK | RCT | 287 [334] | RHZS: 38; RHZE: 39 | RHZS: 33%; RHZE: 38% | — | RHZ+E/S | 4RH | U-INH | PC | Culture (solid) | — |
| Combs [58] | 1990 | US | RCT | 617 | 41 (±15) | 22% | — | 2RHZ±E | 4RH | — | PC | Culture (solid) | 1981–1986 |

(*Continued*)

**Table 2.** (Continued)

| Study | Year | Location | Study design | N assessed[*] | Age (years)[†] | Percent female | Percent HIV+ | Treatment[‡] | | Adherence support | Outcome reported (PC or TTC) | Assessment method[§] | Study dates |
|---|---|---|---|---|---|---|---|---|---|---|---|---|---|
| | | | | | | | | Intensive | Continuation | | | | |
| Lee [59] | 2014 | South Korea | C (r) | 162 | S+: 58 (43–70); C+ 58 (40–70) | S+: 38%; C+: 42% | <1% | 2RHZE[¶] | — | — | Both | Culture (liquid) | 2011–2012 |
| Leung [60] | 2017 | China (Hong Kong) | C (p) | 15,658 [21,414] | 54 (±21) | 36% | <1% | 2RHZ± S/E[¶,**] | NS | — | PC | Smear (NS); culture (NS) | 2006–2010 |
| Musteikienė [61] | 2017 | Lithuania | C (p) | 52 | — | 77% | 0% | 2RHZE | 4RH | IP DOT | PC | Culture (liquid) | 2015–2016 |
| Sajid [62] | 2011 | Pakistan | RCT | 50 | 49 (±18) | 38% | 0% | 2RHZE | — | — | PC | Culture (NS) | 2009–2010 |
| Scott [63][††] | 2017 | US | C (r) | 30,848 | 46 (30–60) | 34% | 6%[‡‡] | 2RHZE[¶,**] | 2RH[¶] | — | PC | Culture (NS) | 2006–2013 |
| TBRC [64] | 1983 | India | RCT | 261 [390] | — | 25% | — | 2RHZS | — | DOT[‖] | PC | Culture (solid) | — |
| Volkmann [65][††] | 2015 | US | C (r) | 60,034 [207,307] | — | 38% | 8% | 2RHZE[¶] | 2RH[¶] | DOT (54%[§§]) | Both | Culture (NS) | 1997–2012 |

BMRC, British Medical Research Council; BTA, British Thoracic Association; C, cohort study (p, prospective; r, retrospective); C+, culture-positive subgroup; DOT, directly observed therapy (values in parentheses indicate duration); E, ethambutol; ECA, East and Central African; FM, fluorescence microscopy; H, isoniazid; HIV−, people living without HIV; HIV+, people living with HIV; HKCS, Hong Kong Chest Service; IP, hospital inpatient; NS, not specified; PC, proportion converted; R, rifampicin; RCT, randomised controlled trial; S, streptomycin; S+, smear-positive subgroup; S. Am., South America; SSA, sub-Saharan Africa; STBS, Singapore Tuberculosis Service; Smoker−, without smoking exposure; Smoker+, with smoking exposure; TB, tuberculosis; TBRC, Tuberculosis Research Centre; TTC, time to conversion; U-INH, urine isoniazid metabolite testing; Z, pyrazinamide; ZN, Ziehl–Neelsen;—, data not available.

[*]Number assessed for outcome. Where demographics were not given for the population assessed for outcome, number in brackets indicates the number for which demographics were stated (which may include drug-resistant TB, non-RHZ-containing regimens, etc.). Where available, per-protocol or efficacy analyses were extracted.

[†]Age is given as median (interquartile range) or mean (± standard deviation) unless otherwise stated.

[‡]In the treatment abbreviations, the number indicates the number of months of treatment, and letters indicate the drugs included. Treatment given 7 days per week unless otherwise stated (5 d = 5 days per week).

[§]For studies performed before 1990, culture type assumed to be solid unless otherwise stated.

[¶]Paper stated treatment was in accordance with national or international guidelines; corresponding regimen shown here. In all other papers, drugs used were explicitly stated.

[‖]Individuals not adherent to DOT (as defined by authors) excluded from analysis.

[**]It is not possible to exclude that some patients were treated with thrice-weekly regimens in these studies.

[††]These studies used the same dataset and overlapping time frames; however, the estimates presented are different. They therefore contribute to different analyses. Two further papers that included overlapping data were excluded to avoid duplication [66,67].

[‡‡]Includes a proportion of patients with unknown HIV status.

[§§]Percentage of study population receiving DOT.

culture medium in Table 5. All 6 studies reporting solid TTC data included only smear-positive patients at baseline. The median TTC ranged from 35 to 49 days, and the mean TTC ranged from 24 to 46 days.

Four studies reported a summary estimate of liquid TTC. Median TTC ranged from 40 to 59 days among baseline smear-positive patients. One study stratified liquid culture TTC by smear status, with a median TTC of 19 days in baseline smear-negative patients compared to 40 days for smear-positive patients.

**Table 3. Number of estimates contributing to proportion converted analysis by time point and microbiological assessment method.**

| Duration of treatment | Smear | Culture | | | Total |
|---|---|---|---|---|---|
| | | Solid | Liquid | Not specified | |
| 1 week | 0 | 4 | 0 | 0 | 4 |
| 2 weeks | 1 | 3 | 1 | 0 | 5 |
| 3 weeks | 1 | 3 | 1 | 1 | 6 |
| 1 month | 4 | 12 | 2 | 2 | 20 |
| 2 months | 10 | 23 | 9 | 4 | 46 |
| 3 months | 1 | 3 | 1 | 2 | 7 |
| 4 months | 1 | 5 | 1 | 1 | 8 |
| **Total** | **18** | **53** | **15** | **10** | **96** |

The proportions of treated patients with DS-TB achieving positive-to-negative culture conversion at weeks 1, 2, and 3 and months 1, 2, 3, and 4 (77 estimates in total) are shown in Fig 3 and Fig 4 for solid and liquid culture methods, respectively.

Most PC data are available at the 2-month time point, where treatment steps down from intensive to maintenance phase. Fig 5 illustrates the proportion of patients achieving culture conversion at 2 weeks and 2 months, restricted to only patients with baseline smear-positive samples.

The proportion of patients treated for DS-TB achieving smear and culture conversion at each time point is summarised in Table 6.

## Animal studies

The search to identify relevant animal studies returned 646 results, from which 3 were found to be relevant at title and abstract sifting; all met inclusion criteria at full-text sifting. A fourth study (Riley et al. [68]) was added from reference and citation checking of the included articles (Fig 6; Table 7).

These 4 papers all describe experimental data from studies in which groups of guinea pigs, used as air samplers, were exposed to air exhausted from dedicated isolation rooms accommodating PTB patients. Three studies evaluated the infectiousness from patients with DS-TB [68–70] to experimental animals, and 1 study considered patients with multi-drug-resistant TB (MDR-TB) [71]. None of the experiments were designed to evaluate temporal changes in

**Table 4. Studies reporting a summary measure of time to smear conversion\*.**

| Study | Year | Country | N | Sampling frequency | Smear stain | Median TTC, days | Mean TTC, days | TTC spread (measure) |
|---|---|---|---|---|---|---|---|---|
| Kennedy [45] | 1996 | Tanzania | 81 | Monthly | FM | — | 55 | 30–120 (range) |
| Dominguez-Castellano [31] | 2003 | Spain | 95 | Weekly | ZN | 20 | — | ±2 (SE) |
| Telzak [56] | 1997 | US | 77 | Weekly | ZN | — | 33 | ±6.2 (SE) |
| Long [46] | 2003 | Canada | 32 | Daily to day 14 then 2×/week† | FM | — | 46 | ±27.9 (SD) |
| Rathored [49] | 2012 | India | 50 | Weekly | ZN | — | 29 | ±0.7 (SD) |
| Méchaï [47] | 2016 | France | 30 | 2×/week‡ | NS | 27 | — | 14–56 (IQR) |

FM, fluorescence microscopy; IQR, interquartile range; N, number assessed for outcome; NS, not stated; SD, standard deviation; SE, standard error; TTC, time to conversion; ZN, Ziehl–Neelsen.

\*Where authors reported data in months, these have been converted to days to aid comparisons (using 30.4 days per month).

†For 14 prospectively studied patients; for 18 retrospectively studied patients sampling was weekly (mean of 1.6 per week) throughout treatment.

‡On days 7–9 then on 3 days every 2 weeks.

| Study | Events | Total | | Proportion | 95%-CI |
|---|---|---|---|---|---|
| **2 weeks** | | | | | |
| Long (2003) | 3 | 32 | | 0.09 | [0.03; 0.24] |
| **3 weeks** | | | | | |
| Long (2003) | 8 | 32 | | 0.25 | [0.13; 0.42] |
| **1 month** | | | | | |
| ECA/BMRC (1983) | 201 | 670 | | 0.30 | [0.27; 0.34] |
| Long (2003) | 11 | 32 | | 0.34 | [0.20; 0.52] |
| Dawson (2009) | 11 | 30 | | 0.37 | [0.22; 0.54] |
| Dlugovitzky (2006) | 6 | 10 | | 0.60 | [0.31; 0.83] |
| **Random effects model** | | **742** | | **0.33** | **[0.25; 0.42]** |
| $I^2 = 33\%$, $\tau^2 = 0.0027$, $p = 0.215$ | | | | | |
| **2 months** | | | | | |
| Dawson (2015) (a) | 24 | 43 | | 0.56 | [0.41; 0.70] |
| Dlugovitzky (2006) | 6 | 10 | | 0.60 | [0.31; 0.83] |
| Stoffel (2014) | 108 | 148 | | 0.73 | [0.65; 0.79] |
| Pheiffer (2008) | 80 | 105 | | 0.76 | [0.67; 0.83] |
| Long (2003) | 25 | 32 | | 0.78 | [0.61; 0.89] |
| Grandjean (2015) | 375 | 467 | | 0.80 | [0.76; 0.84] |
| ECA/BMRC (1983) | 572 | 681 | | 0.84 | [0.81; 0.87] |
| Leung (2017) | 4317 | 4978 | | 0.87 | [0.86; 0.88] |
| Singla (2003) | 391 | 434 | | 0.90 | [0.87; 0.93] |
| Abal (2005) | 309 | 339 | | 0.91 | [0.88; 0.94] |
| **Random effects model** | | **7237** | | **0.82** | **[0.78; 0.86]** |
| $I^2 = 89\%$, $\tau^2 = 0.0048$, $p < 0.001$ | | | | | |
| **3 months** | | | | | |
| Leung (2017) | 4698 | 4978 | | 0.94 | [0.94; 0.95] |
| **4 months** | | | | | |
| Dlugovitzky (2006) | 10 | 10 | | 1.00 | [0.72; 1.00] |

0    0.2    0.4    0.6    0.8    1

**Fig 2. Proportion of patients with baseline smear-positive, drug-susceptible tuberculosis achieving smear conversion at specified time points during effective treatment.** BMRC, British Medical Research Council; CI, confidence interval; ECA, East and Central African.

infectiousness during TB treatment; however, Dharmadhikari et al. re-analysed 5 previous experiments with this objective [71]. As such, the available data are opportunistically available and incomplete, and it is not possible to determine the magnitude of incompleteness. Due to considerable methodological and analytical differences between studies, aggregation of findings was not possible. A brief narrative summary of each individual paper is provided in S2 Appendix.

Overall, the 3 papers evaluating infectiousness of air from patients with DS-TB suggested that those on treatment were less infectious to guinea pigs than untreated patients, although a small number of infections did occur during effective treatment. There are no data that describe how infectiousness changed over time after treatment initiation. It was not possible to

**Table 5. Studies reporting a summary measure of time to solid or liquid culture conversion[*].**

| Study | Year | Country | Smear+ | Sampling frequency | Solid culture | | | Liquid culture | |
|---|---|---|---|---|---|---|---|---|---|
| | | | | | N | Median TTC, days (IQR) | Mean TTC, days (SD) | N | Median TTC, days (IQR) |
| Conde [24] | 2009 | Brazil | 100% | Weekly | 72 | 49 | — | — | — |
| Rathored [49] | 2012 | India | 100% | Every 2 weeks | 50 | — | 24 ± 0.7 | — | — |
| Lee [59] | 2014 | South Korea | 100% | Monthly after 2 weeks | — | — | — | 61 | 40 (28–61) |
| | | | 0% | | | | | 101 | 19 (1–41) |
| Kanda [44] | 2015 | Japan | 100% | Every 2 weeks | 86 | 39 (25–55) | — | — | — |
| Dawson [27] (2015a) | 2015 | South Africa, Tanzania | 100% | Weekly | 54 | 35 | — | 56 | 56 |
| Dawson [28] (2015b) | 2015 | South Africa | 100% | Weekly | 36 | 43 (29–52) | 46 ± 27.9 | 38 | 59 (36–63) |
| Conde [25] | 2016 | Brazil | 100% | Weekly | 45 | 41 (35–48) | — | 36 | 52 (41–59) |

N, number assessed for outcome; IQR, interquartile range; SD, standard deviation; TTC, time to conversion;—, data not available.

[*]Where authors reported data in months, these have been converted to days to aid comparisons (using 30.4 days per month).

disaggregate guinea pig infections occurring from patients with drug-resistant TB on or off effective treatment in the paper by Dharmadhikari et al. It is not possible from these data to draw conclusions about the role of effective treatment in reducing infectiveness of TB, nor about the time frame over which any reduction in infectiousness occurs.

## Discussion

This systematic review and these meta-analyses show that after 2 weeks of effective treatment, 95% and 97% of baseline smear-positive patients demonstrated persistently positive solid and liquid cultures, respectively. This directly challenges guidance that IPC measures can be relaxed following 2 weeks of effective treatment. Despite an effective drug regimen for DS-TB, approximately 1 in 5 baseline smear-positive patients did not achieve smear conversion by 2 months of treatment, decreasing to 1 in 20 by 3 months. It is important for clinicians and programmes to appreciate this significant tail of individuals who, despite receiving effective therapy, remain smear positive. On completion of the 2-month intensive treatment phase, 12% and 41% of baseline smear-positive patients remained solid and liquid culture positive, respectively.

Identification of people with active TB and initiation of effective treatment are vital components of TB IPC, mediated through what is conventionally perceived as a rapid reduction in the infectiousness of treated patients. As part of the evidence gathering for the 2019 WHO guidelines on TB IPC [12], we undertook this systematic review and meta-analysis to investigate the dynamics of sputum sterilisation after initiation of effective treatment.

To our knowledge, this is the first systematic review drawing together the data on the dynamics of sputum sterilisation during effective TB treatment. Strengths of this study include a comprehensive search strategy identifying a large number of studies. We were careful to ensure that all individuals included in the review were receiving an effective drug regimen to avoid confounding by ineffective treatment, and included several outcome assessment methods. A further literature search performed prior to publication did not identify any significant new bodies of work addressing conversion dynamics, or studies that would significantly impact the findings of this review.

The core of this analysis relates to culture conversion as this is an assessment of the presence of viable Mtb in respiratory samples. We included data on smear conversion as the most

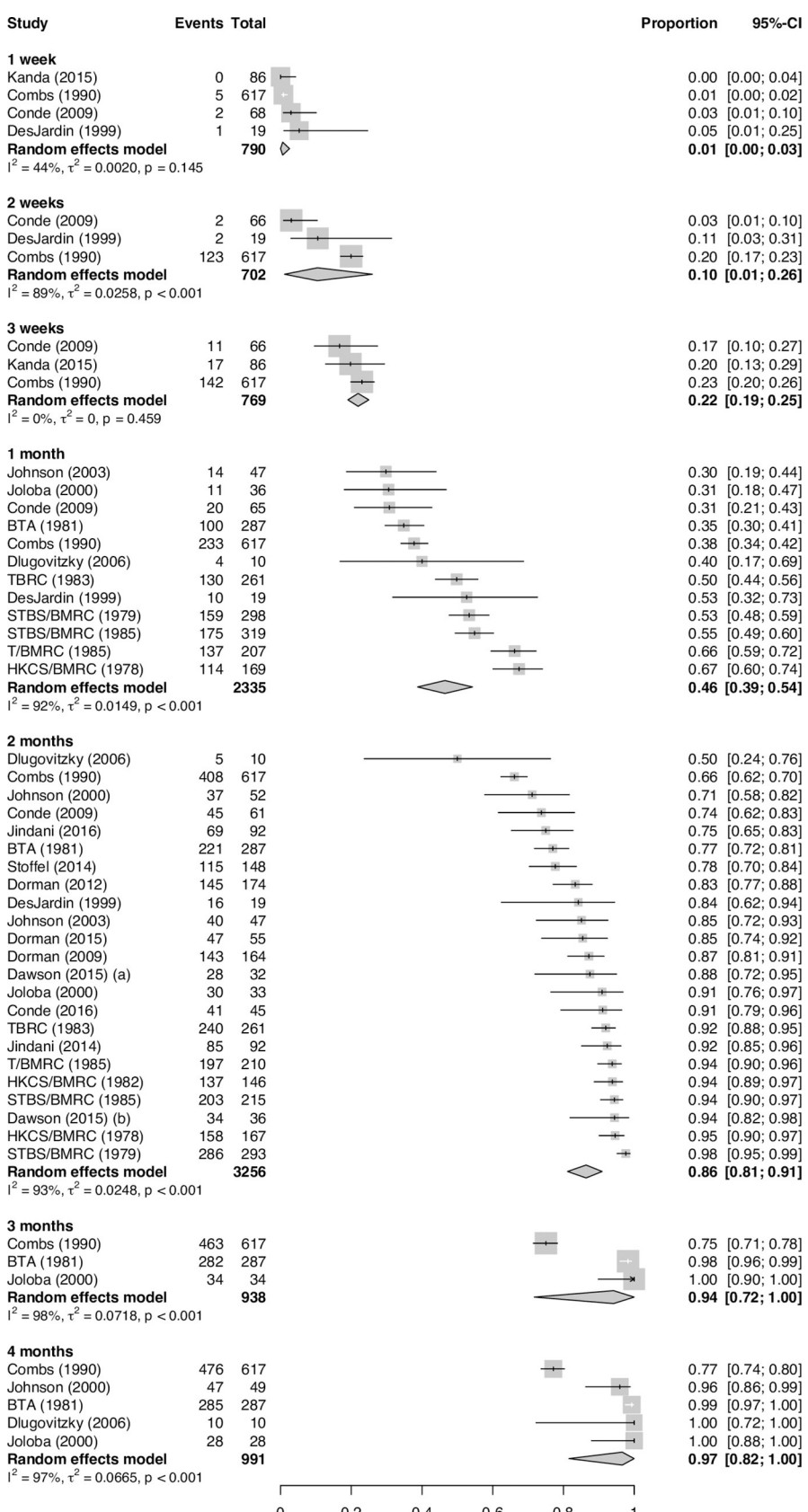

| Study | Events | Total | Proportion | 95%-CI |
|---|---|---|---|---|
| **1 week** | | | | |
| Kanda (2015) | 0 | 86 | 0.00 | [0.00; 0.04] |
| Combs (1990) | 5 | 617 | 0.01 | [0.00; 0.02] |
| Conde (2009) | 2 | 68 | 0.03 | [0.01; 0.10] |
| DesJardin (1999) | 1 | 19 | 0.05 | [0.01; 0.25] |
| **Random effects model** | | **790** | **0.01** | **[0.00; 0.03]** |
| $I^2 = 44\%$, $\tau^2 = 0.0020$, $p = 0.145$ | | | | |
| **2 weeks** | | | | |
| Conde (2009) | 2 | 66 | 0.03 | [0.01; 0.10] |
| DesJardin (1999) | 2 | 19 | 0.11 | [0.03; 0.31] |
| Combs (1990) | 123 | 617 | 0.20 | [0.17; 0.23] |
| **Random effects model** | | **702** | **0.10** | **[0.01; 0.26]** |
| $I^2 = 89\%$, $\tau^2 = 0.0258$, $p < 0.001$ | | | | |
| **3 weeks** | | | | |
| Conde (2009) | 11 | 66 | 0.17 | [0.10; 0.27] |
| Kanda (2015) | 17 | 86 | 0.20 | [0.13; 0.29] |
| Combs (1990) | 142 | 617 | 0.23 | [0.20; 0.26] |
| **Random effects model** | | **769** | **0.22** | **[0.19; 0.25]** |
| $I^2 = 0\%$, $\tau^2 = 0$, $p = 0.459$ | | | | |
| **1 month** | | | | |
| Johnson (2003) | 14 | 47 | 0.30 | [0.19; 0.44] |
| Joloba (2000) | 11 | 36 | 0.31 | [0.18; 0.47] |
| Conde (2009) | 20 | 65 | 0.31 | [0.21; 0.43] |
| BTA (1981) | 100 | 287 | 0.35 | [0.30; 0.41] |
| Combs (1990) | 233 | 617 | 0.38 | [0.34; 0.42] |
| Dlugovitzky (2006) | 4 | 10 | 0.40 | [0.17; 0.69] |
| TBRC (1983) | 130 | 261 | 0.50 | [0.44; 0.56] |
| DesJardin (1999) | 10 | 19 | 0.53 | [0.32; 0.73] |
| STBS/BMRC (1979) | 159 | 298 | 0.53 | [0.48; 0.59] |
| STBS/BMRC (1985) | 175 | 319 | 0.55 | [0.49; 0.60] |
| T/BMRC (1985) | 137 | 207 | 0.66 | [0.59; 0.72] |
| HKCS/BMRC (1978) | 114 | 169 | 0.67 | [0.60; 0.74] |
| **Random effects model** | | **2335** | **0.46** | **[0.39; 0.54]** |
| $I^2 = 92\%$, $\tau^2 = 0.0149$, $p < 0.001$ | | | | |
| **2 months** | | | | |
| Dlugovitzky (2006) | 5 | 10 | 0.50 | [0.24; 0.76] |
| Combs (1990) | 408 | 617 | 0.66 | [0.62; 0.70] |
| Johnson (2000) | 37 | 52 | 0.71 | [0.58; 0.82] |
| Conde (2009) | 45 | 61 | 0.74 | [0.62; 0.83] |
| Jindani (2016) | 69 | 92 | 0.75 | [0.65; 0.83] |
| BTA (1981) | 221 | 287 | 0.77 | [0.72; 0.81] |
| Stoffel (2014) | 115 | 148 | 0.78 | [0.70; 0.84] |
| Dorman (2012) | 145 | 174 | 0.83 | [0.77; 0.88] |
| DesJardin (1999) | 16 | 19 | 0.84 | [0.62; 0.94] |
| Johnson (2003) | 40 | 47 | 0.85 | [0.72; 0.93] |
| Dorman (2015) | 47 | 55 | 0.85 | [0.74; 0.92] |
| Dorman (2009) | 143 | 164 | 0.87 | [0.81; 0.91] |
| Dawson (2015) (a) | 28 | 32 | 0.88 | [0.72; 0.95] |
| Joloba (2000) | 30 | 33 | 0.91 | [0.76; 0.97] |
| Conde (2016) | 41 | 45 | 0.91 | [0.79; 0.96] |
| TBRC (1983) | 240 | 261 | 0.92 | [0.88; 0.95] |
| Jindani (2014) | 85 | 92 | 0.92 | [0.85; 0.96] |
| T/BMRC (1985) | 197 | 210 | 0.94 | [0.90; 0.96] |
| HKCS/BMRC (1982) | 137 | 146 | 0.94 | [0.89; 0.97] |
| STBS/BMRC (1985) | 203 | 215 | 0.94 | [0.90; 0.97] |
| Dawson (2015) (b) | 34 | 36 | 0.94 | [0.82; 0.98] |
| HKCS/BMRC (1978) | 158 | 167 | 0.95 | [0.90; 0.97] |
| STBS/BMRC (1979) | 286 | 293 | 0.98 | [0.95; 0.99] |
| **Random effects model** | | **3256** | **0.86** | **[0.81; 0.91]** |
| $I^2 = 93\%$, $\tau^2 = 0.0248$, $p < 0.001$ | | | | |
| **3 months** | | | | |
| Combs (1990) | 463 | 617 | 0.75 | [0.71; 0.78] |
| BTA (1981) | 282 | 287 | 0.98 | [0.96; 0.99] |
| Joloba (2000) | 34 | 34 | 1.00 | [0.90; 1.00] |
| **Random effects model** | | **938** | **0.94** | **[0.72; 1.00]** |
| $I^2 = 98\%$, $\tau^2 = 0.0718$, $p < 0.001$ | | | | |
| **4 months** | | | | |
| Combs (1990) | 476 | 617 | 0.77 | [0.74; 0.80] |
| Johnson (2000) | 47 | 49 | 0.96 | [0.86; 0.99] |
| BTA (1981) | 285 | 287 | 0.99 | [0.97; 1.00] |
| Dlugovitzky (2006) | 10 | 10 | 1.00 | [0.72; 1.00] |
| Joloba (2000) | 28 | 28 | 1.00 | [0.88; 1.00] |
| **Random effects model** | | **991** | **0.97** | **[0.82; 1.00]** |
| $I^2 = 97\%$, $\tau^2 = 0.0665$, $p < 0.001$ | | | | |

**Fig 3. Proportion of baseline culture-positive patients receiving effective treatment for drug-susceptible tuberculosis achieving solid culture conversion at specified time points.** BMRC, British Medical Research Council; BTA, British Thoracic Association; CI, confidence interval; HKCS, Hong Kong Chest Service; STBS, Singapore Tuberculosis Service; T, Tanzania; TBRC, Tuberculosis Research Centre.

frequently used microbiological test used for response to therapy; however, smear positivity of treated patients may not reflect presence of viable *Mtb*. The evaluation of TB transmission from human patients to experimental animals provides an additional perspective.

Whilst presence of *Mtb* in respiratory secretions is a necessary component of infectiousness, assessment of the capacity to detect *Mtb* in spontaneously produced sputum does not directly quantify infectiousness. Infectiousness also depends on the environment, a mechanism for propagation (e.g., cough dynamics), and recipient factors. This is a limitation of this review

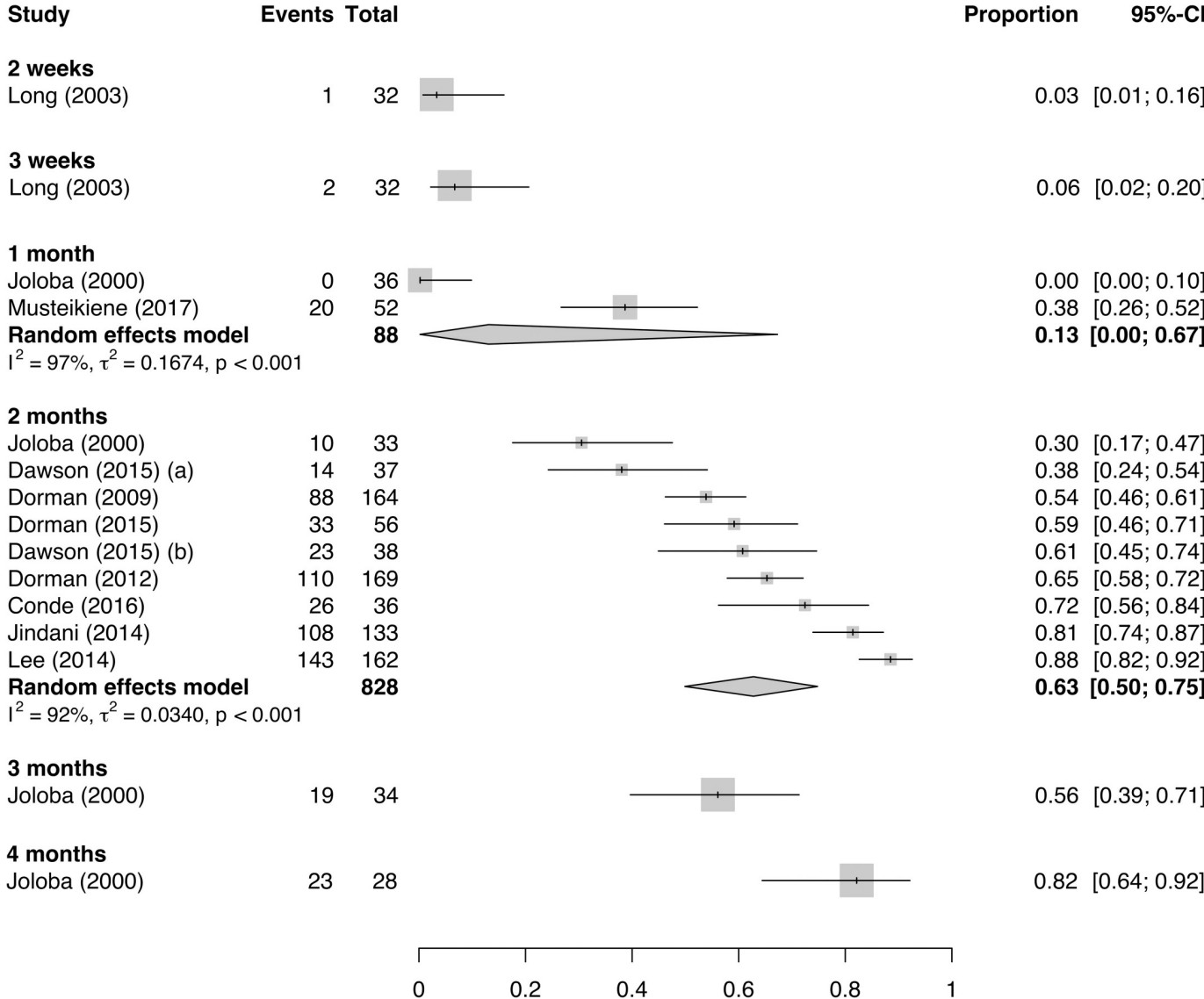

**Fig 4. Proportion of baseline culture-positive patients receiving effective treatment for drug-susceptible tuberculosis achieving liquid culture conversion at specified time points.** CI, confidence interval.

| Study | Events | Total | | Proportion | 95%-CI |
|---|---|---|---|---|---|
| **2 weeks** | | | | | |
| **Liquid culture** | | | | | |
| Long (2003) | 1 | 32 | | 0.03 | [0.01; 0.16] |
| | | | | | |
| **Solid culture** | | | | | |
| Conde (2009) | 2 | 66 | | 0.03 | [0.01; 0.10] |
| DesJardin (1999) | 2 | 19 | | 0.11 | [0.03; 0.31] |
| **Random effects model** | | | | **0.05** | **[0.00; 0.14]** |
| $I^2 = 40\%$, $\tau^2 = 0.0055$, p = 0.197 | | | | | |
| | | | | | |
| **2 months** | | | | | |
| **Liquid culture** | | | | | |
| Joloba (2000) | 10 | 33 | | 0.30 | [0.17; 0.47] |
| Dawson (2015) (a) | 14 | 37 | | 0.38 | [0.24; 0.54] |
| Dorman (2009) | 88 | 164 | | 0.54 | [0.46; 0.61] |
| Dorman (2015) | 33 | 56 | | 0.59 | [0.46; 0.71] |
| Dawson (2015) (b) | 23 | 38 | | 0.61 | [0.45; 0.74] |
| Dorman (2012) | 110 | 169 | | 0.65 | [0.58; 0.72] |
| Conde (2016) | 26 | 36 | | 0.72 | [0.56; 0.84] |
| Jindani (2014) | 108 | 133 | | 0.81 | [0.74; 0.87] |
| **Random effects model** | | | | **0.59** | **[0.47; 0.70]** |
| $I^2 = 87\%$, $\tau^2 = 0.0213$, p < 0.001 | | | | | |
| | | | | | |
| **Solid culture** | | | | | |
| Dlugovitzky (2006) | 5 | 10 | | 0.50 | [0.24; 0.76] |
| Johnson (2000) | 37 | 52 | | 0.71 | [0.58; 0.82] |
| Conde (2009) | 45 | 61 | | 0.74 | [0.62; 0.83] |
| Jindani (2016) | 69 | 92 | | 0.75 | [0.65; 0.83] |
| Stoffel (2014) | 115 | 148 | | 0.78 | [0.70; 0.84] |
| Dorman (2012) | 145 | 174 | | 0.83 | [0.77; 0.88] |
| DesJardin (1999) | 16 | 19 | | 0.84 | [0.62; 0.94] |
| Johnson (2003) | 40 | 47 | | 0.85 | [0.72; 0.93] |
| Dorman (2015) | 47 | 55 | | 0.85 | [0.74; 0.92] |
| Dorman (2009) | 143 | 164 | | 0.87 | [0.81; 0.91] |
| Dawson (2015) (a) | 28 | 32 | | 0.88 | [0.72; 0.95] |
| Joloba (2000) | 30 | 33 | | 0.91 | [0.76; 0.97] |
| Conde (2016) | 41 | 45 | | 0.91 | [0.79; 0.96] |
| Jindani (2014) | 85 | 92 | | 0.92 | [0.85; 0.96] |
| T/BMRC (1985) | 197 | 210 | | 0.94 | [0.90; 0.96] |
| HKCS/BMRC (1982) | 137 | 146 | | 0.94 | [0.89; 0.97] |
| STBS/BMRC (1985) | 203 | 215 | | 0.94 | [0.90; 0.97] |
| Dawson (2015) (b) | 34 | 36 | | 0.94 | [0.82; 0.98] |
| HKCS/BMRC (1978) | 158 | 167 | | 0.95 | [0.90; 0.97] |
| STBS/BMRC (1979) | 286 | 293 | | 0.98 | [0.95; 0.99] |
| **Random effects model** | | | | **0.88** | **[0.84; 0.92]** |
| $I^2 = 85\%$, $\tau^2 = 0.0140$, p < 0.001 | | | | | |

0    0.2    0.4    0.6    0.8    1

**Fig 5. Proportion of baseline smear-positive patients receiving effective treatment for drug-susceptible tuberculosis, achieving solid culture and liquid culture conversion at 2 weeks and 2 months.** BMRC, British Medical Research Council; CI, confidence interval; ECA, East and Central African; HKCS, Hong Kong Chest Service; STBS, Singapore Tuberculosis Service; T, Tanzania.

**Table 6. Overview of summary estimates derived from random effects meta-analysis for the proportion of patients achieving culture conversion at each time point, by detection method.**

| Duration of treatment | Smear microscopy | | | Solid culture | | | Liquid culture | | |
|---|---|---|---|---|---|---|---|---|---|
| | Proportion (95% CI) | N | $I^2$ | Proportion (95% CI) | N | $I^2$ | Proportion (95% CI) | N | $I^2$ |
| 1 week | — | 0 | — | 0.01 (0.00–0.03) | 4 | 44% | - | 0 | — |
| 2 weeks | 0.09 (0.03–0.24) | 1 | — | 0.10 (0.01–0.26) | 3 | — | 0.03 (0.01–0.16) | 1 | — |
| 3 weeks | 0.25 (0.13–0.42) | 1 | — | 0.22 (0.19–0.25) | 3 | — | 0.06 (0.02–0.20) | 1 | — |
| 1 month | 0.33 (0.25–0.42) | 4 | 32% | 0.46 (0.39–0.54) | 12 | 91% | 0.17 (0.09–0.25) | 1 | — |
| 2 months | 0.82 (0.78–0.86) | 10 | 89% | 0.86 (0.81–0.91) | 23 | 93% | 0.63 (0.50–0.75) | 9 | 92% |
| 3 months | 0.94 (0.94–0.95) | 1 | — | 0.94 (0.72–1.00) | 3 | — | 0.56 (0.39–0.71) | 1 | — |
| 4 months | 1.00 (0.72–1.00) | 1 | — | 0.97 (0.82–1.00) | 5 | 97% | 0.82 (0.64–0.92) | 1 | — |

CI, confidence interval; N, number of estimates.

with regard to public health policy. Cough frequency diminishes significantly in the first few weeks of effective TB treatment. This likely plays an important role in reducing transmission risk, but was outside the scope of this review [4,11]. We also excluded studies reporting human TST conversion, which could provide a 'real world' assessment of infectiousness. To our knowledge, however, there are no studies reporting TST conversion rates attributable to specific time points or time periods following treatment initiation. We also did not include quantitative evaluations of sputum mycobacterial load after treatment initiation, for example from studies of early bactericidal activity. Declining sputum mycobacterial load after initiation of treatment could provide evidence for declining infectiousness; however, without culture conversion, the time at which an individual becomes non-infectious cannot be determined.

Another limitation for the interpretation of the presented estimates is the high level of observed heterogeneity between studies, resulting in a lack of precision in the presented estimates. This is a consequence of the diverse patient populations sampled across a high number of included studies. There is no 'typical' TB cohort. Patients vary widely in terms of sex, age, exposure history, disease duration and severity (including smear status and presence or absence of lung cavitation), *Mtb* strain lineage, nutritional status, profile of comorbidities such as HIV and diabetes, and tolerance of and adherence to treatment. Many of these covariates are independently associated with time to sputum conversion [31,56]. This leads to important variation in time to culture conversion between individuals, which is magnified by aggregation of patients into study populations. Where possible we conducted subgroup analyses to explore observed heterogeneity; however, few studies disaggregated results by key covariates, and heterogeneity remained high, even within subgroups.

In our analyses, we found that liquid culture remained positive for longer than solid culture and that smears assessed by fluorescence microscopy staining were positive for longer than those assessed by Ziehl–Neelsen staining. This is unsurprising, given liquid culture and fluorescence microscopy staining are considered more sensitive detection methods [72,73]. However, observed differences may also reflect imbalances in confounding covariates across subgroups. Lack of disaggregation by conversion-predicting covariates such as HIV status, baseline smear status, and the presence of cavitating lung disease prevented any meaningful

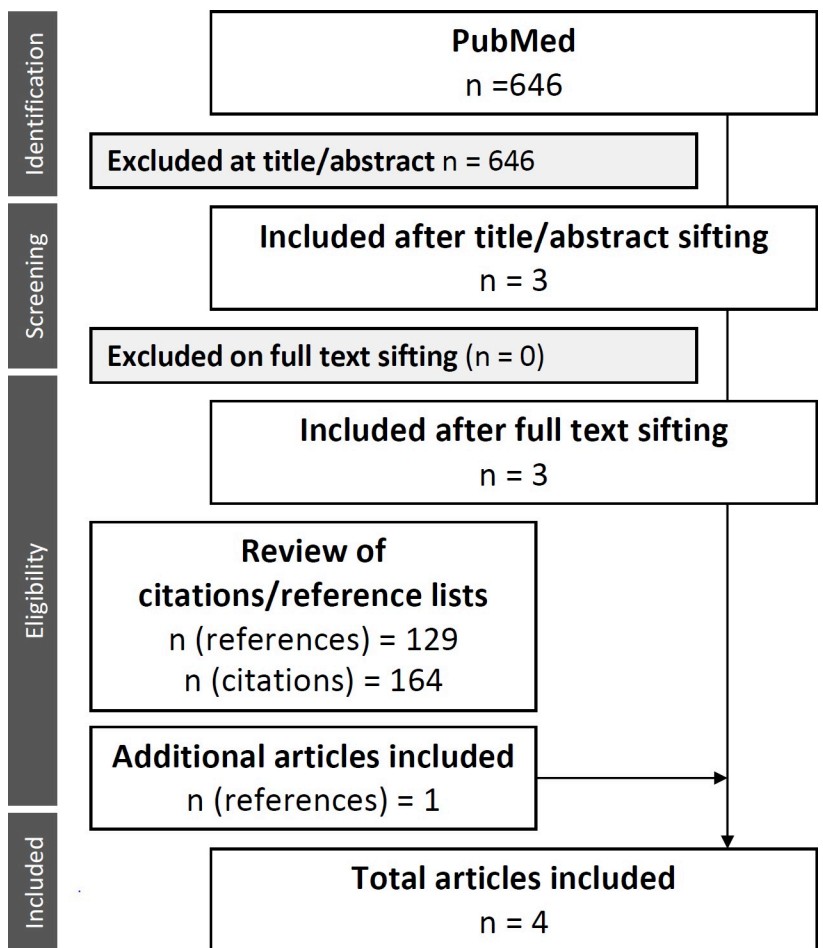

**Fig 6. Flow diagram showing process of identification of animal studies for review.** n, number of studies.

attempt to explore potential confounding. Point estimates of TTC are expected to be influenced by the frequency of sampling; however, this parameter was often not reported. These limitations should be recognised when inferring a relationship between detection method and conversion estimates.

The proportion of estimated new cases successfully treated for MDR-TB globally was 55% at the time of this work [1]. The scope of the review was therefore restricted to patients with proven DS-TB in an effort to ensure all included patients were receiving effective treatment.

**Table 7. Studies evaluated for data on infectiousness from TB patients to animals during TB treatment.**

| Study | Year | Drug susceptibility of population | Untreated patients (*N* or percent) | Treated patients (*N* or percent) | Treatment duration |
|---|---|---|---|---|---|
| Riley [68] | 1962 | DS | 67 | 40 | Not clear |
| Escombe [69] | 2007 | Mixed* | 17% of patient-days | 83% of patient-days | Not clear |
| Escombe [70] | 2008 | Mixed* | | | Not clear |
| Dharmadhikari [71] | 2014 | MDR/XDR | Variable (across 5 studies) | | Not clear |

DS, drug-susceptible; MDR, multi-drug-resistant; *N*, number assessed; XDR, extensively drug-resistant.

*Included individuals with drug-susceptible TB and those with drug-resistant TB.

Our findings are therefore not generalisable to MDR-TB. The restriction of the review to DS-TB also resulted in the exclusion of many articles, including all search results reported only in abstract form, because drug susceptibility testing was not performed or not reported, or because conversion estimates were not disaggregated by drug susceptibility testing results.

Adherence is critical to effective treatment. Many studies reported using directly observed therapy and some included other mechanisms for adherence support, but very few reported adherence data. The possibility of incomplete adherence further limits the interpretation of our findings. Consequently, our analyses should be considered analogous to an intention-to-treat rather than an on-treatment protocol.

Beyond the indirect measure of sputum conversion time alone, we sought to explore when TB treatment renders patients non-infectious by reviewing studies using experimental animals as air samplers. In these studies, air extracted from isolation rooms housing TB patients is exhausted over susceptible guinea pigs, exposing them to infectious droplet nuclei. Modification of patient conditions, including the use of effective treatment, can be exploited to infer the effect of interventions on infectiousness, assessed by serial tuberculin skin testing of exposed animals. These studies suggest that patients receiving effective TB treatment were less infectious than those not receiving such treatment; however, the temporal dynamics of infectiousness could not be elicited from the identified studies, individually or when combined.

This systematic review was performed to inform the WHO Department of Global TB Programme GDG of the evidence available to answer the question 'How does the infectiousness of TB patients (ability to excrete viable bacteria and sustain transmission) change after having started effective TB treatment?' Our summary estimates of culture conversion suggest that the majority of patients excrete viable bacilli at 2 weeks of treatment, and many continue to do so for several months. What this means for the 2-week 'rule' cannot be determined by these analyses, as we did not evaluate other factors that are also important for TB transmission. These are important gaps to address for evidence-based TB IPC guidance. Integrating the findings of this review with data on cough dynamics, the impact of environmental factors such as air changes per hour, and varying host susceptibility through a modelling approach could provide a more comprehensive assessment of TB transmission during treatment. Studies of TST conversions of experimental animals, with the objective of evaluating transmission dynamics after TB treatment initiation, would also provide new insight.

Viable *Mtb* persists in sputum for months after initiation of TB treatment. Understanding the implications of this for TB transmission from patients on treatment necessitates a comprehensive evaluation including not only the persistence of *Mtb* in sputum, but also the contribution of other host and pathogen factors, including cough dynamics.

## Supporting information

**S1 PRISMA Checklist. Table containing the PRISMA 2009 checklist items and referencing their position in the paper.**
(DOC)

**S1 Appendix. Description and outcome of updated literature search, run on 22 November 2020.**
(DOCX)

**S2 Appendix. Narrative review of animal studies that review the infectiousness of patients with tuberculosis.**
(DOCX)

**S1 Fig. Proportion of baseline culture-positive patients receiving effective treatment for drug-susceptible TB achieving culture conversion (culture method unspecified) at 3 weeks and at 1, 2, 3, and 4 months.**
(TIF)

**S1 Form. Pilot-tested Microsoft Excel data extraction form.**
(XLSX)

**S1 Table. Details of databases searched and terms used for original search, run on 1 December 2017.**
(DOCX)

**S2 Table. Details of databases searched and terms used for secondary search, run on 20 February 2018.**
(DOCX)

**S3 Table. Details of databases searched and terms used for animal studies search, run on 27 March 2018.**
(DOCX)

**S4 Table. Summary of quality scores for included primary research studies, assessed using an adapted National Institutes of Health tool for case series.**
(DOCX)

**S5 Table. Studies reporting a summary measure of time to culture conversion where the culture type was not specified.**
(DOCX)

## Acknowledgments

Particular thanks are extended to Maria Krutikov and Mengyun Liu for their assistance with duplicate sifting and data extraction, and the support from Lice González-Angulo and Fuad Mirzayev from the WHO.

## Author Contributions

**Conceptualization:** Claire J. Calderwood, James P. Wilson, Katherine L. Fielding, David A. J. Moore.

**Formal analysis:** Claire J. Calderwood, James P. Wilson, Rebecca C. Harris, Aaron S. Karat, Raoul Mansukhani, David A. J. Moore.

**Investigation:** Claire J. Calderwood, James P. Wilson, Katherine L. Fielding, Rebecca C. Harris, Aaron S. Karat, Raoul Mansukhani, Jane Falconer, Malin Bergstrom, Sarah M. Johnson, Nicky McCreesh, Edward J. M. Monk, Jasantha Odayar, Peter J. Scott, Sarah A. Stokes, Hannah Theodorou, David A. J. Moore.

**Methodology:** Claire J. Calderwood, James P. Wilson, Katherine L. Fielding, Rebecca C. Harris, Jane Falconer, David A. J. Moore.

**Project administration:** Katherine L. Fielding, David A. J. Moore.

**Resources:** Jane Falconer.

**Supervision:** Katherine L. Fielding, David A. J. Moore.

**Visualization:** Raoul Mansukhani.

**Writing – original draft:** Claire J. Calderwood, James P. Wilson, Aaron S. Karat, David A. J. Moore.

**Writing – review & editing:** Claire J. Calderwood, James P. Wilson, Katherine L. Fielding, Rebecca C. Harris, Aaron S. Karat, Raoul Mansukhani, Jane Falconer, Malin Bergstrom, Sarah M. Johnson, Nicky McCreesh, Edward J. M. Monk, Jasantha Odayar, Peter J. Scott, Sarah A. Stokes, Hannah Theodorou, David A. J. Moore.

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
