## [Editor Report · Decision Letter 0]

11 Jul 2020

Dear Dr Wilson, 

Thank you for submitting your manuscript entitled "Sputum conversion during tuberculosis treatment: systematic review and meta-analysis" for consideration by PLOS Medicine.

Your manuscript has now been evaluated by the PLOS Medicine editorial staff as well as by an academic editor with relevant expertise and I am writing to let you know that we would like to send your submission out for external peer review.

Kind regards,

Thomas J McBride, PhD,

PLOS Medicine

---

## [Decision Letter · Decision Letter 1]

16 Sep 2020

Dear Dr. Wilson,

Thank you very much for submitting your manuscript "Sputum conversion during tuberculosis treatment: systematic review and meta-analysis" (PMEDICINE-D-20-03202R1) for consideration at PLOS Medicine. 

[LINK]

In light of these reviews, I am afraid that we will not be able to accept the manuscript for publication in the journal in its current form, but we would like to consider a revised version that addresses the reviewers' and editors' comments. Obviously we cannot make any decision about publication until we have seen the revised manuscript and your response, and we plan to seek re-review by one or more of the reviewers. 

We expect to receive your revised manuscript by Oct 07 2020 11:59PM. Please email us (plosmedicine@plos.org) if you have any questions or concerns.

We look forward to receiving your revised manuscript. 

Sincerely,

Emma Veitch, PhD

PLOS Medicine

On behalf of Tom McBride, PhD

plosmedicine.org

*Please structure your abstract using the PLOS Medicine headings (Background, Methods and Findings, Conclusions).

*We recommend ensuring that the abstract (ideally Methods and Findings section) includes a brief note about any key limitations of the study's methodology.

*At this stage, we ask that you include a short, non-technical Author Summary of your research to make findings accessible to a wide audience that includes both scientists and non-scientists. The Author Summary should immediately follow the Abstract in your revised manuscript. This text is subject to editorial change and should be distinct from the scientific abstract. Please see our author guidelines for more information: https://journals.plos.org/plosmedicine/s/revising-your-manuscript#loc-author-summary

*The methods section of the paper notes that the systematic review was conducted per PRISMA guidelines, in general PRISMA is described (or was conceived) as a tool to assist reporting, not necessarily conduct - it might be an idea to rephrase this as the SR being reported per PRISMA (rather than conducted). We'd also suggest including a completed PRISMA checklist as supporting information alongside the paper.

*The discussion section at the moment does not seem to include much of an explicit acknowledgement of main limitations of the SR methods (or evidence base), some issues relating to generalisability and clinical implications are discussed but not really "signposted" as limitations of the SR per se. We'd recommend adding something to more explicitly describe this. 

*One reviewer queries the definition of what data to include in the SR - the editors (including academic editor) acknowledged that it would not be possible to entirely address this issue without totally redoing the SR, so we'd just recommend including a discussion on this point in relation to study limitations (if any). 

Comments from the reviewers:

Reviewer #1: This review looked at culture conversion in smear positive pulmonary tuberculosis patients after several durations of effective anti tuberculosis therapy.

The findings indicate that the majority of patients (97% by liquid culture) are still culture positive after two weeks of effective treatment.

These findings are of great interest to infection control policy makers.

I have several questions for the authors to consider

1. Would it be possible to do a subgroup analysis or HIV infected vs HIV uninfected patients?

2. Was there any difference in culture conversion time between patients receiving and not receiving ethambutol?

3. Was any consideration given to looking at culture conversion times in smear negative culture positive patients?

4. Was it possible to look at the effect of smear grade or initial time to culture on culture conversion at 2 weeks?

Reviewer #2: I confine my remarks to statistical aspects of this paper. These were well done and I recommend publication

peter Flom

Reviewer #3: This is a well-written paper, with a clear research question and appropriate methodology and conclusions. The topic is of significant clinical relevance, and would be a significant addition to the current knowledge. Regarding statistics I do not have relevant experience to assess the accuracy. 

Reviewer #4: Summary 

This is a systematic review commissioned by WHO TB GDG with the aim of summarising how long MTB can be detected in sputum in population of adults initiating treatment for pulmonary tuberculosis, as a proxy for "potential infectiousness". The authors look at 2 summary measure types (time to conversion from positive to negative, and proportion converting to negative at given timepoints) for two method types (microscopy smear and culture). In addition, a narrative review of animal studies is included. Unsurprisingly the authors find that a majority of PTB patients remain smear and culture positive after 2 weeks treatment, but with substantial heterogeneity between studies in all outcomes. The authors found that the literature is not well enough reported (ie disaggregated) to allow any quantitative analysis of this heterogeneity (eg meta-regression). They report that the animal study data does not provide any estimates of infectiousness change over time on treatment and hint at suspected bias from incomplete data (probably there is more detail given in the supplement). The authors conclude that "potential infectiousness" extends well into TB treatment, but are careful to note that "What this means for TB transmission and the two-week 'rule' cannot be determined by these analyses". 

Comments

The review is very well written, and meets its objectives using sound analyses which are well presented. 

It meets all PRISMA 2009 checklist criteria (assume PRISMA item 8 & 11 are in the supplement), except there is no mention of systematic review registration number or indication if a review protocol exists / if/ where it can be accessed (PRISMA 2 & 5).

The lit search was performed 2.5 years ago.

Any possible major weaknesses stem from the terms of reference for the review (commissioned by WHO) rather than the manuscript itself I think. E.g. the two week rule comes from a review by Rouillon in Tubercle 1976 and was based on the absence of reported cases of TST conversion in household contacts of PTB patients after the index patients were discharged on treatment (which ignores the survivor bias from these contacts likely having already been exposed and not converted). I wondered why the animal data was reviewed but not this type of epidemiological data. I also thought the EBA pharmacodynamics based on serial quantification of bacilli in sputum could have been relevant to the review question (not least because it is much better powered to assess the heterogeneity issue). However as stated above the review is very clear and meets the stated objectives completely so this is just an aside. 

Minor comments

Typo in table 1: 

Any study in a not reporting data on participants with confirmed, drug susceptible…

Line 167: Thirty-six studies reported data on culture conversion; 9 using liquid culture and 23 using solid (8 studies reported both).

I had to read this a few times to understand why couldn't make 36 from 9, 23, and 8 - it only makes sense when you read the next line. Would be clearer if edited maybe.

Reviewer #5: The authors did a great work in analysing the sputum conversion time after treatment onset. The study design, meta-analyses, and write-up of the study is well done. However, I am quite worried about the conclusion, due to the smaller sample size obtained from the included studies. I am also worried about the inclusion of the animal components when it did not add up anything to the conclusion or to strengthen the study. Other concerns and comments are found in the included/attached PDF.

[LINK]

---

## [Decision Letter · Decision Letter 2]

20 Nov 2020

Dear Dr. Wilson,

Thank you very much for submitting your revised manuscript "Dynamics of sputum conversion during effective tuberculosis treatment: a systematic review and meta-analysis" (PMEDICINE-D-20-03202R2) for consideration at PLOS Medicine. 

Your revision was evaluated by a senior editor and discussed among all the editors here. It was also discussed with the academic editor and sent to two of the original reviewers. The reviews are appended at the bottom of this email and any accompanying reviewer attachments can be seen via the link below:

[LINK]

I am afraid that we still will not be able to accept the manuscript for publication in the journal in its current form, but we would like to consider a further revised version that addresses the editors' remaining comments. Obviously we cannot make any decision about publication until we have seen the revised manuscript and your response, and we may seek re-review by one or more of the reviewers. 

We expect to receive your revised manuscript by Dec 04 2020 11:59PM. Please email us (plosmedicine@plos.org) if you have any questions or concerns.

We look forward to receiving your revised manuscript. 

Sincerely,

Thomas McBride, PhD

Senior Editor 

PLOS Medicine

plosmedicine.org

1- Thank you for providing your PRISMA checklist. Please replace the page numbers with paragraph numbers per section (e.g. "Methods, paragraph 1"), since the page numbers of the final published paper may be different from the page numbers in the current manuscript.

2- Data statement: Is it more accurate to say “Data are available from the primary studies, all of which are published.”

3- In the Abstract Methods and Findings, please provide the databases searched, the beginning and end dates of your search, types of study designs included, eligibility criteria, and synthesis/appraisal methods.

4- Please begin the Abstract Conclusions with “This systematic review found that most patients remained…”

5- The last sentence of the Abstract Conclusions should be replaced or modified to suggest what is needed to understand the other components of infectiousness and integrate the findings of this study.

6- Author summary, line 58: instead of “not strong”, perhaps “challenged”?

7- The searches are nearly 3 years old, please update to the present time.

8- Discussion, line 388, please remove “the best”.

9- Please present and organize the Discussion as follows: a short, clear summary of the article's findings; what the study adds to existing research and where and why the results may differ from previous research; strengths and limitations of the study; implications and next steps for research, clinical practice, and/or public policy; one-paragraph conclusion.

10- The contents of S3 Table read “Table E3”

11- S1 Form is empty, can you provide the completed form?

Comments from the reviewers:

Reviewer #1: Thank you for addressing the questions I raised in the review. These have all been answered. 

Reviewer #5: I thank the authors for such a good work. I am convinced this will be a good addition to science, public health and the literature. Congratulations on such a good work!

[LINK]

---

## [Editor Report · Decision Letter 3]

3 Feb 2021

Dear Dr. Wilson,

Thank you very much for re-submitting your manuscript "Dynamics of sputum conversion during effective tuberculosis treatment: a systematic review and meta-analysis" (PMEDICINE-D-20-03202R3) for review by PLOS Medicine.

I have discussed the paper with my colleagues and the academic editor and it was also seen again by reviewers 1 and 5. I am pleased to say that provided the remaining editorial and production issues are dealt with we are planning to accept the paper for publication in the journal.

[LINK]

We look forward to receiving the revised manuscript by Feb 10 2021 11:59PM.   

Sincerely,

Raffaella

Dr Raffaella Bosurgi MSc, PhD

Executive Editor, PLOS Medicine

rbosurgi@plos.org

https://twitter.com/raffi74

Remote based in London, UK

PLOS

Requests from Editors:

Comments from Reviewers:

[LINK]

---

## [Editor Report · Decision Letter 4]

15 Feb 2021

Dear Dr Wilson, 

On behalf of my colleagues and the Academic Editor Claudia M Denkinger I am pleased to inform you that we have agreed to publish your manuscript "Dynamics of sputum conversion during effective tuberculosis treatment: a systematic review and meta-analysis" (PMEDICINE-D-20-03202R4) in PLOS Medicine.

PRESS

Sincerely, 

Dr Raffaella Bosurgi 

Executive Editor 

PLOS Medicine